

# Skin microbiota variation in Indian families

Renuka Potbhare[1], Ameeta RaviKumar[2], Eveliina Munukka[3,4],
Leo Lahti[5] and Richa Ashma[1]

[1] Department of Zoology, Savitribai Phule Pune University, Pune, Maharashtra, India
[2] Institute of Bioinformatics and Biotechnology, Savitribai Phule Pune University, Pune, Maharashtra, India
[3] Turku Clinical Microbiome Bank, Department of Clinical Microbiology, Turku University Hospital, Turku, Finland
[4] Institute of Biomedicine, University of Turku, Turku, Finland
[5] Department of Computing, Faculty of Technology, University of Turku, Turku, Finland

Corresponding authors
Leo Lahti, leo.lahti@utu.fi
Richa Ashma,
richaashma@unipune.ac.in

## ABSTRACT

**Background:** In India, joint families often encompass members spanning multiple generations cohabiting in the same household, thereby sharing the same ethnicity, genetics, dietary habits, lifestyles, and other living conditions. Such an extended family provides a unique opportunity to evaluate the effect of genetics and other confounding factors like geographical location, diet and age on the skin microbiota within and between families across three generations.

**Methods:** The present study involved seventy-two individuals from fifteen families from two geographical regions of Maharashtra, India. The 16S rRNA sequencing of V3–V4 regions was performed and the generated taxonomic profiles were used for downstream analysis.

**Results:** Our study highlights a significant difference in community composition (beta diversity) between families (PERMANOVA; $p = 0.001$) and geographical locations ($p = 0.001$). We observed geographical location-wise differences in the relative abundances *Staphylococcus* in the families from Pune (Wilcoxon test, $p = 0.007$), and *Bacillus* in the Ahmednagar families (Wilcoxon test, $p = 0.004$). When within and between-family comparisons of skin microbiota composition were carried out between different generations (G1–G2, G2–G3, and G1–G3); we observed skin microbiota tended to be more similar within than between families but this difference was not significant.

**Conclusion:** This study underscores the diversity and commonalities in skin microbiota composition within and between families. Our result suggests that geographical location is significantly associated with the genus composition of skin microbiota, which is quantitatively unique for a family and likely explained by co-habitation.

## INTRODUCTION

Skin is an epithelial interface mediating the interaction between the internal and external body environment, and it provides a first line of defence against toxins and invasion of various pathogens (*Gallo, 2017*; *Coates et al., 2019*; *Williams et al., 1998*). Human skin also offers a niche for diverse microbial communities of bacteria, fungi, viruses, *etc.*, collectively known as skin microbiota (*Byrd, Belkaid & Segre, 2018*). These microbes are classified as resident or transient species based on their survival on the human skin surface (*Grice et al., 2008*). Resident species live longer on the skin, and are not harmful. They mainly belong to *Propionibacterium*, *Corynebacterium*, and *Staphylococcus* genera (*Scholz & Kilian, 2016*). However, depending on the skin's micro-environment, these resident bacteria may have a positive or opportunistic impact (*Findley & Grice, 2014*). For instance, in optimal environments, some bacteria (*e.g., Staphylococcus aureus*) are beneficial to skin health as they enhance immune responses and inhibit the colonization of detrimental pathogens. Whereas, under conditions, such as, changes in the temperature, pH, and disruption of skin barrier function, the same beneficial resident bacteria (*e.g., Staphylococcus aureus*) may become opportunistic causing infections or other dermatological diseases (*Findley & Grice, 2014*). Transient species like *Bacillus sp.*, *Staphylococcus aureus*, and *Pseudomonas aeruginosa* are short-term residents as they are easily influenced by changes in the physio-chemical properties of the skin (*Bojar & Holland, 2002*).

Microbiota composition is notably diverse on healthy skin and is affected by multiple endogenous and exogenous factors such as age (*Chaudhari et al., 2020*; *Kim et al., 2019*), sex (*Byrd, Belkaid & Segre, 2018*; *Li et al., 2019*), diet (*Salem et al., 2018*), geography (*Ross et al., 2018*), and environment (*Peng & Biswas, 2020*; *Lehtimäki et al., 2017*). Current evidence supports the transmission of skin microbiota from mother to child depending on the type of delivery, *i.e.*, babies born vaginally exhibit a microbiota composition similar to that of their mother's birth canal (*Yao et al., 2021*). At the same time, infants delivered by cesarean section acquire bacterial communities resembling the mother's skin surfaces (*Yao et al., 2021*; *Dominguez-Bello et al., 2010*). Maternal health, breastfeeding, close contact, and other factors also play a vital role in shaping the skin microbiota of an infant (*LaTuga, Stuebe & Seed, 2014*; *Rapin et al., 2023*). Likewise, skin type (dry, moist, sebaceous), skin site, and skin parameters such as pH, sebum levels, and the number of hair follicles and glands contribute in shaping skin microbiota communities (*Oh et al., 2014*; *Cho & Eom, 2021*). Additionally, the host lifestyle, hygiene (*Riverain-Gillet et al., 2020*), and use of cosmetics (*Wallen-Russell, 2018*; *Salverda et al., 2013*) influence the abundance and composition of microbial communities that inhabit the skin.

Despite these endogenous and exogenous factors, microbiota varies between individuals and can also be influenced by genetics (*Si et al., 2015*; *Skowron et al., 2021*). However, understanding the role of genetics remains unclear. It has not been interrogated thoroughly as the skin microbiota studies have primarily focused on associations with human cohabitation (*Ross, Doxey & Neufeld, 2017*), human-pet association (*Song et al., 2013*), captivity, and indoor environment (*Lax et al., 2014*). Research about the skin microbiota associations among genetically related individuals has been limited.

Joint or extended families, a feature of Indian society since ancient times (*Karve, 1965*; *Mullaiti, 1995*); which comprises a group of multi-generational people, living together under one roof, preparing food at a common hearth, *i.e.*, sharing similar dietary habits, hygiene, and house environmental conditions (*Karve, 1965*; *Chadda & Deb, 2013*). Moreover, family membership is also a proxy for genetic relations between many family members. Thus, the study of skin microbiota similarity in such multi-generational families provides a unique opportunity to study the associations of genetic relatedness with skin microbiota composition across three generations. The present study evaluates links between family membership and thus also genetic relatedness on the skin microbiota composition. This is especially important as the family or genetic relations have not been evaluated on skin microbiota (reports are available on the gut microbiome), in the joint Indian families. Hence, the present study analysed 72 individuals from fifteen families wherein three generations live together (cohabitation) under the same roof, sharing food and other household items. This type of family structure helps to control for the effect of confounding factors, such as diet, sex, and age. We further evaluated the effect of confounding factors on skin microbiota diversity and composition and compared families to identify family-specific microbiota variation. Additionally, we also investigated the associations linked to genetic (family) relations to understand skin microbiome variation within and between families across three generations.

## MATERIALS AND METHODS

### Study design and subject enrolment

Portions of this text were previously published as part of a preprint (https://www.biorxiv.org/content/10.1101/2023.12.09.570904v1). Seventy-two volunteers comprising fifteen families were enrolled in the present study from two districts of Maharashtra, India, viz, Pune (altitude: 1,840 ft; longitude: 73°51′19.26″E; latitude: 18°31′10.45″N) and Ahmednagar (altitude: 2,129 ft; longitude: 74°44′58.53″E; latitude: 19°5′42.75″N). The family members had similar dietary habits, vegetarian or mixed, with the exception of two families (Table 1). Family members were categorised into three generations by age, generation-1, grandmother or grandfather (G1, age 65–91 yrs.), generation-2, mother or father (G2, 41–63 yrs.), and generation-3, daughter or son respectively (G3, 13–30 yrs.). The volunteers fulfilled the following conditions and criteria: a) Each family must comprise individuals from above mentioned three generations. b) All members of the same family shared same household. c) All volunteers were self-identified as healthy. The volunteers were instructed to avoid using deodorants, skin ointments, and soaps-12–15 hrs prior the sampling, shaving of axilla at least 2 days before sampling, tobacco, smoking, and alcohol consumption, and certain food items such as onion, garlic, chilies. The demographics of the fifteen enrolled families are represented in Fig. 1. The map outline and state boundary data obtained from GADM version 4.1 (http://www.gadm.org).

Volunteers were informed about the study goals and provided a questionnaire to obtain information about lifestyle and medication history. The questionnaire covered the history of dermatological disease, use of antibiotics (5–6 months prior the sampling), long-term

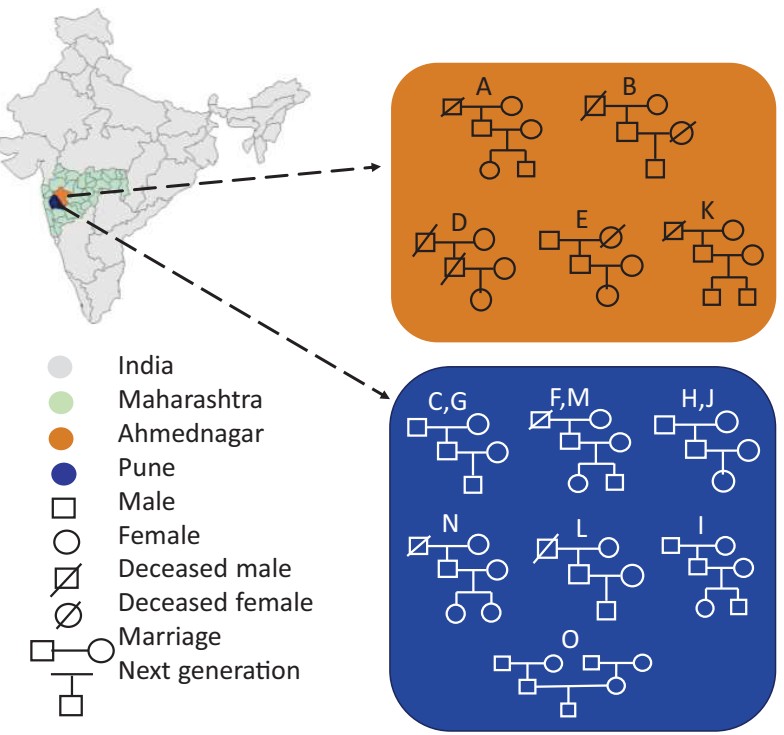

**Figure 1 Demographics of the enrolled families (A-O) represents geography-wise family distribution and pedigree structure across three generations.** The map outline and state boundary data obtained from GADM.

medication history, dietary habits, level of physical activity, *etc.* (Table S1). The written informed consent was obtained from volunteers prior the sample collection. The experiments involved in the study were approved by an appropriate institutional human ethics committee of Savitribai Phule Pune University (Letter No. SPPU/IEC/2019/57). The statistical power for this experiment calculated using RNASeqPower calculator for common skin phyla (Firmicutes and Proteobacteria) is >80%, by setting alpha level 0.05 and effect size 2.

## Sample collection and microbial DNA extraction

The screening of families were done prior the sample collection and volunteers were informed in advance about the sampling schedule. All members of the same family were sampled simultaneously, on the same day between 10 am to 12 pm. This schedule mitigated the impact of temperature change, ensuring that the observed result was not attributed to environmental fluctuations during the day thus enhanced the rigor of the study controlling environmental biasness. It was challenging to find families with all three generations living together under the same roof in cities like Pune and Ahmednagar, limiting the sample size available for this study. The most enrolled families had only one grandparent (G1) or a single sibling (G3). Altogether fifteen families were sampled in 2–3 days. The sweat swabs were collected from the moist skin site, *i.e.*, the axilla/armpit, as it is easily accessible and less exposed skin surface, allowing to capture of a greater number of

**Table 1 Information on the fifteen enrolled families with confounding factors.**

| Family | Members | Generation | | | Diet | | Sex | | Location | | Age | | |
|---|---|---|---|---|---|---|---|---|---|---|---|---|---|
| | | G1 | G2 | G3 | Vegetarian | Mixed | Male | Female | Pune | Ahmednagar | Elderly | Middle age | Adult |
| (n = 15) | (n = 72) | (n = 32) | (n = 28) | (n = 21) | (n = 57) | (n = 15) | (n = 33) | (n = 39) | (n = 52) | (n = 20) | (n = 25) | (n = 26) | (n = 21) |
| A | 5 | 1 | 2 | 2 | 5 | 0 | 2 | 3 | 0 | 5 | 1 | 2 | 2 |
| B | 3 | 1 | 1 | 1 | 3 | 0 | 2 | 1 | 0 | 3 | 1 | 1 | 1 |
| C | 5 | 2 | 2 | 1 | 5 | 0 | 3 | 2 | 5 | 0 | 2 | 2 | 1 |
| D | 3 | 1 | 1 | 1 | 3 | 0 | 0 | 3 | 0 | 3 | 1 | 1 | 1 |
| E | 4 | 1 | 2 | 1 | 0 | 4 | 2 | 2 | 0 | 4 | 2 | 1 | 1 |
| F | 5 | 1 | 2 | 2 | 1 | 4 | 2 | 3 | 5 | 0 | 1 | 2 | 2 |
| G | 5 | 2 | 2 | 1 | 5 | 0 | 3 | 2 | 5 | 0 | 2 | 2 | 1 |
| H | 5 | 2 | 2 | 1 | 5 | 0 | 2 | 3 | 5 | 0 | 2 | 2 | 1 |
| I | 6 | 2 | 2 | 2 | 6 | 0 | 3 | 3 | 6 | 0 | 2 | 2 | 2 |
| J | 5 | 2 | 2 | 1 | 0 | 5 | 2 | 3 | 5 | 0 | 2 | 2 | 1 |
| K | 5 | 1 | 2 | 2 | 5 | 0 | 3 | 2 | 0 | 5 | 2 | 1 | 2 |
| L | 4 | 1 | 2 | 1 | 4 | 0 | 2 | 2 | 4 | 0 | 1 | 2 | 1 |
| M | 5 | 1 | 2 | 2 | 5 | 0 | 2 | 3 | 5 | 0 | 1 | 2 | 2 |
| N | 5 | 1 | 2 | 2 | 3 | 2 | 1 | 4 | 5 | 0 | 1 | 2 | 2 |
| O | 7 | 4 | 2 | 1 | 7 | 0 | 4 | 3 | 7 | 0 | 4 | 2 | 1 |

resident bacteria. The samples were collected in triplicate using sterile cotton swabs moistened with PBS solution (1×). Cotton buds were scrubbed multiple times in upward and downward directions to capture the axillary microbiota. To prevent the sample degradation, the sweat samples were stored immediately in 4 °C cooler at the collection site and transported to the laboratory. The genomic DNA was extracted from the samples using methods described previously (*Potbhare et al., 2022*) and stored at −80 °C until sequencing. In brief, the bacterial cells were lysed using lysis buffer (0.5 M EDTA, 0.5 M Tris-HCl, 0.1 v/v Triton X, 8% sucrose), followed by double purification method described in *Sambrook, Fritsch & Maniatis (1989)*. The mixture was precipitated by washing several times with 100% and 70% ethanol. The resulting nucleic acid mixture was confirmed for DNA by loading on agarose gel and assessed for quality and quantity using a nanodrop spectrophotometer at 260 and 280 nm absorbance. The DNA was pooled together from triplicate samples before sequencing to avoid sequence artefacts associated with low biomass.

## Library preparation and *16S rRNA* gene sequencing

We selected V3–V4 regions of *16S rRNA* as they provide a comprehensive assessment of microbial diversity across samples and have higher resolution for lower-rank taxa (*Bukin et al., 2019*). It was also reported that the V3–V4 region has higher accuracy for the expected abundance of recognized taxa compared to the V1–V3 region (*Castelino et al., 2017*; *Potbhare et al., 2022*). The universal prokaryotic primers of V3–V4 regions were used
to amplify the *16S rRNA* gene. Polymerase chain reactions (PCR) were performed using KAPA HiFi HotStart Ready Mix® (KAPA Biosystems, Boston, MA, USA) with the following thermal parameters: initial denaturation at 95 °C for 3 min, followed by 25 cycles of 95 °C for 30 s; 55 °C for the 30 s; and 72 °C for 30 s. The resulting amplicons were purified with AMPure XP beads using PureLink™ PCR purification kit® (Invitrogen, Waltham, MA, USA). Further, dual indices and adaptors were linked to the amplicon using the Nextera XT Index Kit® (Illumina, San Diego, CA, USA) following the thermal cycler program: initial denaturation at 95 °C for 3 min, followed by eight cycles of 95 °C for the 30 s; 55 °C for the 30 s; and 72 °C for 5 min. These adaptor-ligated libraries were purified with AMPure XP beads using PureLink™ PCR purification kit® (Invitrogen, Waltham, MA, USA). The quantitation of PCR products was performed using Qubit™ dsDNA HS Assay Kit® (Invitrogen, Waltham, MA, USA) on a 2.0 fluorometer (Life Technologies, Waltham, MA, USA). These products were pooled into equal molar proportions and sequenced on Illumina MiSeq V2 standard flow cell (Illumina, San Diego, CA, USA) for 2 * 300 bp pair-end chemistry according to the manufacturer's instructions. A 5% PhiX control (Illumina, San Diego, CA, USA) along with positive (DNA sample extracted from the healthy gut) and negative controls (MilliQ water sample) were included in the final run to rule out the possibility of contamination from the experimental materials.

## Bioinformatics

The bacterial *16S rRNA* gene sequences in FASTQ format were aligned using Illumina BaseSpace toolkit. The primer and adapter sequences were trimmed from the raw sequences. The sequences were filtered after trimming the 3′ end with the quality score (Q) 30 using "FastQ toolkit" application of Illumina BaseSpace and DADA2. This eliminated PCR-generated chimeras, contaminants, and low-quality reads from the sequences. In order to generate the OTU table, RefSeq RDP 16S v3 May 2018 DADA2 was used for taxonomic identification (*Callahan et al., 2016*). Altogether 5,710,132 reads, ranging between 30,281–134,873 reads with an average ~80,000 reads per sample were generated. The amplicon sequence reads were then grouped into 1,070 OTUs based on <98% sequence similarity cutoff. Then the data was imported into *TreeSummarizedExperiment* data container in R to organize the taxonomic profiling data and associated metadata on samples and features (v. 2.6.0) (*Huang et al., 2021*).

## Statistical analysis

We quantified alpha diversity with Shannon index, and associated this with multiple covariates including diet, age, sex, geographical location, and family. Further, dissimilarities in taxonomic composition between individuals (beta diversity) was calculated with Permutational Multivariate Analysis of Variance (PERMANOVA) with 999 permutations using Bray-Curtis dissimilarity (*Anderson, 2001*). Principal coordinates analysis (PCoA) based on Bray-Curtis dissimilarity was used to visualize the distribution of taxonomic composition among the study population. The beta diversity measurements were calculated using the *vegan* R package (*Oksanen et al., 2020*). The *p*-values <=0.05, <=0.1 considered significant and borderline significant, respectively. The differential

abundance analysis using ancombc2 (analysis of compositions of microbiomes with bias correction 2) was performed to identify bacterial taxa that were significantly different in families and location (*Lin & Peddada, 2024*). The *p*-values were calculated using Kruskal-Wallis test, genera with *p*-value < 0.05 were considered differentially abundant. The pairwise multiple comparison was subsequently carried out using Dunn *post-hoc* test and adjusted with Benjamini–Hochberg method.

In order to investigate the similarity in the microbiota composition among family members across three generations, we compared individuals from G1–G2 (grandparent-parent), G2–G3 (parent-children), and G1–G3 (grandparent-children) and calculated within and between family (dis)similarities using Bray-Curtis index. For this analysis, we randomly selected one member from each generation of the same family and calculated pair-wise within family similarity. For between family differences, we randomly selected members of each generation from different families and calculated pair-wise between family (dis)similarity. The pairwise comparison of G2 members of the same family was also carried out, to check the association of sex on skin microbiota when diet, age and location were the same. Further, familywise inter-generational analysis was performed by selecting two members per generation (G1–G2 or G2–G3) using Bray-Curtis index. We further extended G2–G3 (parent-child) analysis, and separately compared father-child and mother-child skin microbiota similarity to obtain baseline (dis)similarity. The *p*-values were adjusted for multiple testing with Benjamini-Hochberg False Discovery Rate (FDR) correction for all the analyses. Comparisons with FDR < 0.05 were considered significant, and with FDR < 0.1-borderline significant. Further, we divided the Bray-Curtis dissimilarities into three groups as low (<=0.25), intermediate (0.25–0.75), and high (>=0.75) while comparing families. We limited all comparisons within the same city, age, and sex to control the potential association of these factors with skin microbiota composition.

The analyses were carried out at the genus level unless otherwise mentioned. All analyses and visualizations were carried out in R (version 4.2.2). The primary R packages were *mia* (1.6.0) and *miaViz* (1.6.0) (*Ernst, Borman & Lahti, 2022*; *Ernst et al., 2022*).

# RESULTS

## Taxonomic profiling of skin microbiota

Skin microbiota observed in our study comprised a total of 36 phyla, 84 classes, 191 orders, 326 families, and 1,071 genera. Most of these observed groups were rare and observed only in very few samples and at a low abundance. Of the 36 phyla observed across 15 families, three most dominant phyla were *Firmicutes* (prevalence 100%, mean relative abundance 73%), *Proteobacteria* (97.2%, 23.8%), and *Actinobacteria* (90.2%, 3.2%) having >1% prevalence and a detection threshold of 0.1% (Table 2). We filtered the observed 1,071 genera by setting a detection threshold of 0.1% and prevalence >1% for the inclusion of maximum genera in the analysis. The set prevalence threshold resulted in the inclusion of 73 genera for the analysis. The remaining genera were grouped as 'other'. Out of 73 genera the most dominant six genera with highest mean abundance were *Staphylococcus* (prevalence 100%, mean relative abundance 50.9%), *Bacillus* (87.5%, 16.3%), *Pseudomonas*

(61.1%, 9.2%), *Anaerococcus* (38.9%, 1.7%), *Corynebacterium* (73.6%, 1%), and *Dermabacter* (40.2%, 0.4%) (Table 3).

## Skin microbiome diversity analysis for various confounding factors

To investigate the overall diversity and abundance of microbiota communities in 72 samples, we estimated typical measures of alpha diversity in a sample, richness (observed richness), evenness (Pielou's evenness), and diversity (Shannon index) at genus level. The alpha diversity measures were controlled for confounding factors including age, diet, sex, and geographical location. We observed a borderline association of diet (Wilcoxon test, $p = 0.08$) and geographical location (Wilcoxon test, $p = 0.08$) on skin microbiota. The differences were not significant for age (Kruskal-Wallis test, adult & middle age, $p = 0.4$; adult & elderly, $p = 0.62$; middle age & elderly, $p = 0.92$) and sex (Wilcoxon test, $p = 0.73$) (Fig. 2A).

Beta diversity measures compare genus composition between the analysed samples. We quantified beta diversity using the Bray-Curtis index which calculates dissimilarity between samples based on microbial relative abundances. A high beta diversity index indicates low community similarity, while a low beta diversity index implies a high similarity. We observed a significant difference in the skin microbiota of different families (PERMANOVA; $p = 0.001$) and geographical location ($p = 0.001$). We did not observe a significant difference in taxonomic composition for diet ($p = 0.923$), age ($p = 0.317$), or sex ($p = 0.467$) (Table 4). Individual's skin microbiota similarity based on family and location is further illustrated with principal coordinates analysis (Fig. 2B). Additionally, a distance-based redundancy analysis (dbRDA) with major significant confounding factors (geographical location and family) was performed to highlight these significant differences (explained variance 0%, $p = 0.01$; 27.7%, $p = 0.01$). However, dbRDA did not provide significant differences with diet (0.2%, $p = 0.9$), age (2%, $p = 0.4$), or sex (0.8%, $p = 0.4$), as illustrated in Fig. 2C.

## Family-specific microbiome variation across geographical locations

We observed borderline to significant differences in both alpha and beta diversity for geographical location. Further quantitative analysis based on the relative abundance of the three most prevalent phyla (Table 2) and top six genera (Table 3) across families was carried out (Figs. 3A, 3B). On the most prevalent top six genera, differential abundance analysis (DAA) with ancombc2 at 1% prevalence was performed. This revealed, a high abundance of *Staphylococcus* in the families from Pune (Wilcoxon test, $p = 0.007$), and *Bacillus* in the Ahmednagar families (Wilcoxon test, $p = 0.004$) (Fig. 3C). Using the same analysis, further, significant-difference was also observed in the other twenty-three genera across two geographical locations (Fig. S1). Our result suggests that geographical location is associated with skin microbiota composition.

Additionally, an inter-family comparison was performed to check the possible family-specific microbiome variation. This was carried out using DAA with ancombc2 at 1% prevalence. Our result revealed significant quantitative differences in the genus-level composition between families from both geographical locations (Table 5 and Fig. S2). Our

**Table 2 The most prevalent phyla on skin.** The most prevalent phyla on skin microbiome with the mean relative abundance, median, prevalence (detection threshold = 0.1%, prevalence > 1%), and inter-quantile range.

| Phylum (%) | Mean relative abundance (%) | Median (%) | Prevalence (%) | Inter quartile range (%) |
|---|---|---|---|---|
| *Firmicutes* | 73.0 | 93.5 | 100.0 | 53.90 |
| *Proteobacteria* | 23.8 | 2.6 | 97.2 | 49.30 |
| *Actinobacteria* | 3.2 | 1.2 | 90.2 | 2.42 |

result suggests that there is family-specific skin microbiota composition. Next, we investigated differences between individuals for the most prevalent six genera in the families of the two geographical locations. The relative abundances varied within families (Fig. S3).

Furthermore, as the geographical location was identified as the most important factor controlling microbial compositions, we further checked the contribution of other confounders on each location using dbRDA. We observed borderline to significant differences in the 'family' (as a factor) on Pune ($p = 0.07$) and Ahmednagar samples ($p = 0.001$). The differences were not significant for age, gender and diet ($p >= 0.1$) on the samples from two geographical locations (Table S2). The PCA visualization of samples from two locations is represented in Fig. S4.

## Inter-generational analysis of skin microbiota within and between families

We further performed inter-generational analysis to examine microbiome dissimilarity within and between family members across the three generations. Our analysis did not reveal significant differences in the microbiota composition between generations: G1–G2 (grandparent-parent), G2–G3 (parent-children), and G1–G3 (grandparent-children) for both within and between family comparisons (Kruskal-Wallis test, FDR < 0.1, Figs. 4A–4C). However, we observed that families residing in different geographical locations showed greater variation (Bray-Curtis index >= 0.75) across generations (G1–G2, G2–G3, and G1–G3) (Table S3). These dissimilarities were high in two out of five families from Ahmednagar and three out of 10 families from Pune. Such variation could be due to the association of intrinsic (age and gender) or extrinsic factors (outdoor activities). Further, the pairwise comparison of G2 members of the same family was carried out to assess the association of sex with skin microbiota composition when diet, age group and location were kept constant. The intermediate level of Bray Curtis dissimilarity observed in the families across geographical locations. In few families the differences detected high (three low, five intermediate, five high) (Table S4).

Furthermore, when we conducted a separate analysis of G2–G3 (parent-child), we again did not find significant differences within and between families for father and mother, (Kruskal-Wallis test, FDR < 0.1, Figs. 4D, 4E). On average children exhibited a greater degree of microbiota similarity with their parents than other G2 members. Also, we did not

**Table 3 The most prevalent genera on skin.** The most prevalent genera on skin microbiome with the mean relative abundance, median, prevalence (detection threshold = 0.1%, prevalence > 1%), and inter-quantile range.

| Genus | Mean relative abundance (%) | Median (%) | Prevalence (%) | Inter quartile range (%) |
|---|---|---|---|---|
| *Staphylococcus* | 50.9 | 60.2 | 100.0 | 91.80 |
| *Bacillus* | 16.3 | 0.3 | 87.5 | 12.90 |
| *Pseudomonas* | 9.2 | 0.1 | 61.1 | 4.98 |
| *Anaerococcus* | 1.7 | 0.0 | 38.9 | 0.26 |
| *Corynebacterium* | 1.0 | 0.2 | 73.6 | 0.91 |
| *Dermabacter* | 0.4 | 0.0 | 40.2 | 0.61 |

**A.**

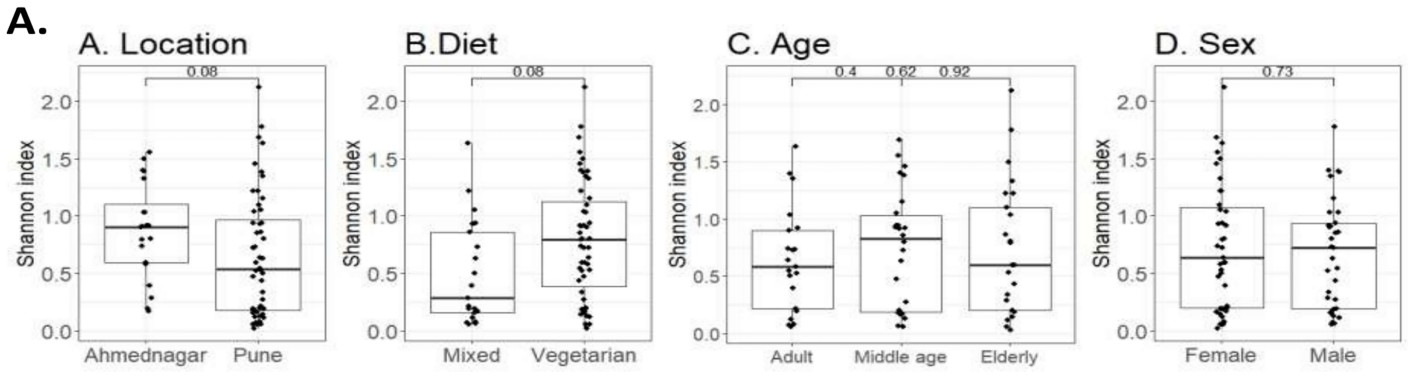

**B.** **C.**

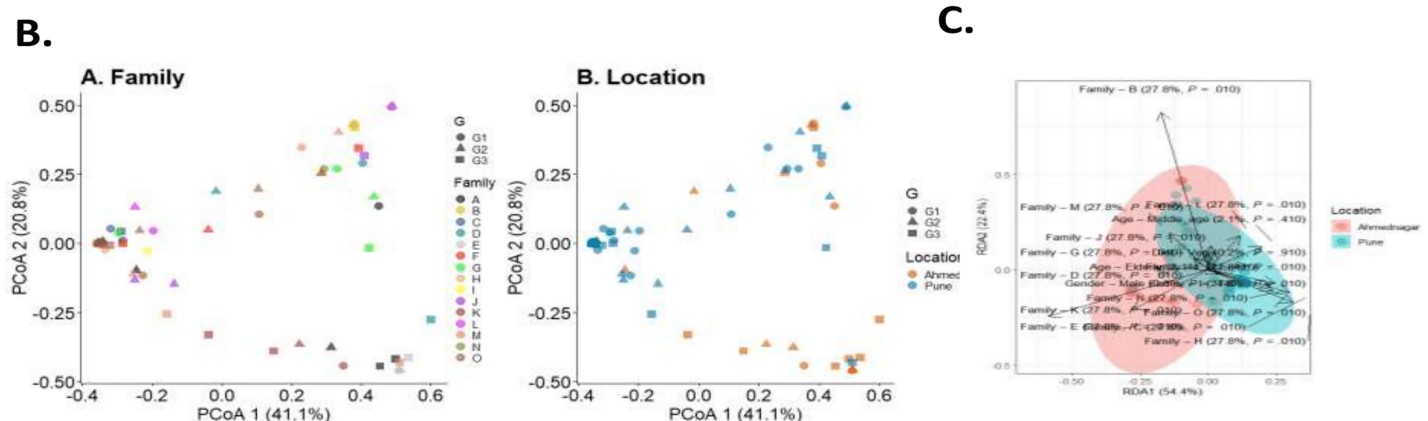

**Figure 2 Skin microbiota diversity analysis for various confounding factors.** (A) Alpha diversity (Shannon index) variation of the skin microbiota and its associations with (A) location (Wilcoxon test, $p = 0.08$); (B) diet (Wilcoxon test, $p = 0.08$); (C) age (Kruskal-Wallis test, $p = 0.73$); (D) sex (Wilcoxon test, $p = 0.73$). (B) Dissimilarity of skin microbiota composition illustrated on Principal Coordinates Analysis ordination (PCoA; Bray-Curtis). The shape indicates generation (G1, G2, and G3) and color indicates (A) family; (B) location. (C) Distance-based redundancy (dbRDA) analysis of skin microbiota with major confounding factors (Bray-Curtis dissimilarity), family (27.7%, $p = 0.01$), location (0%, $p = 0.01$), diet (0.2%, $p = 0.9$), age (2%, $p = 0.4$) and sex (0.8%, $p = 0.4$).

observe significant inter-generational differences (G1–G2 and G2–G3) within families (Fig. S5). We also investigated whether children shared a higher taxonomic similarity with their father compared to their mother but the difference was not significant

**Table 4 Beta diversity analysis of skin microbiome.** Association between the skin microbiome taxonomic composition with key covariates (PERMANOVA; Bray-Curtis dissimilarity; Benjamini-Hochberg adjustment for multiple testing).

| Factors | R2 | *p*-value |
|---|---|---|
| Diet | 0.002 | 0.923 |
| Age | 0.020 | 0.317 |
| Sex | 0.008 | 0.467 |
| Location | 0.000 | 0.001*** |
| Family | 0.278 | 0.001*** |

Note:
The *p*-value indicates level of significance <0.001***.

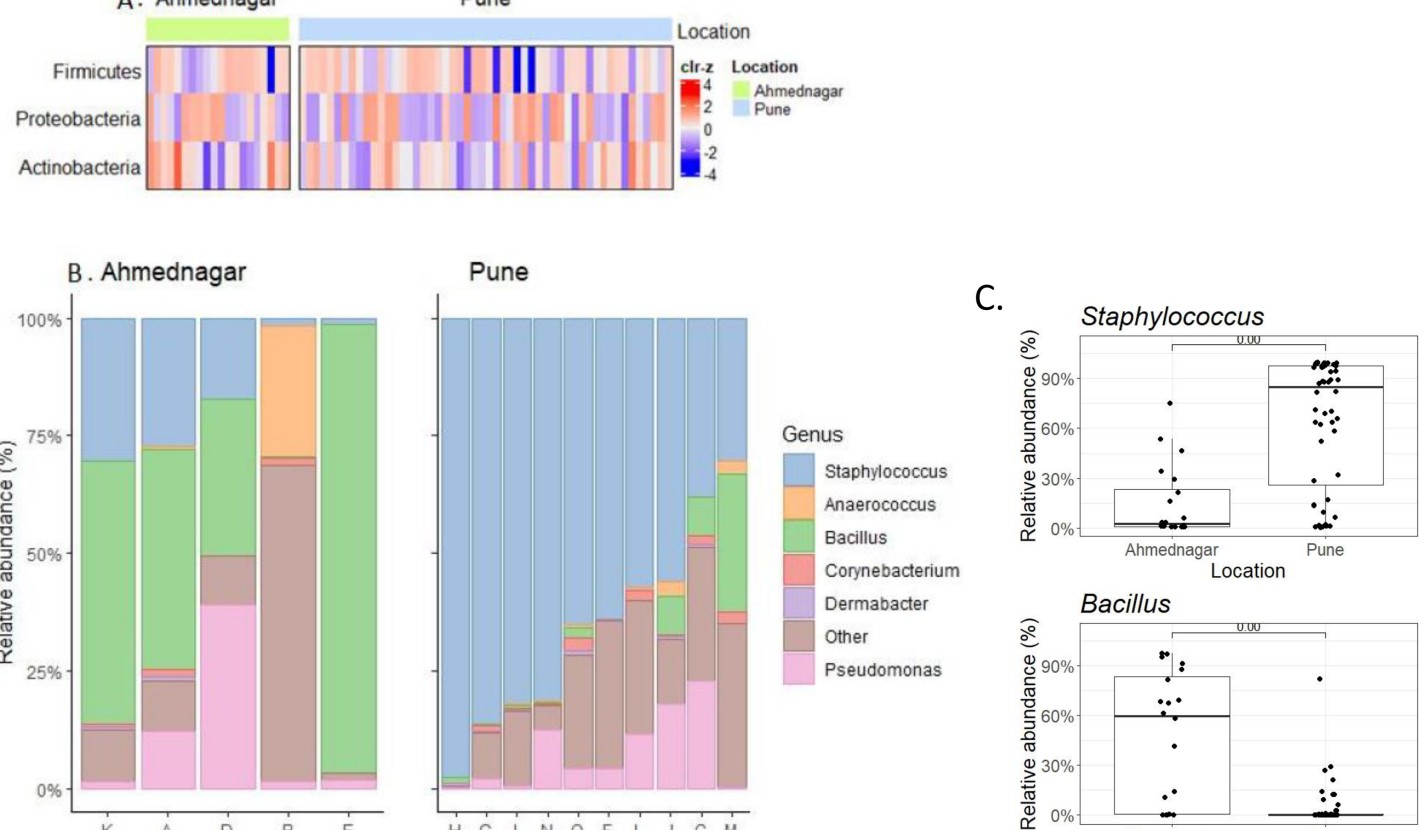

**Figure 3 The most prevalent phyla and genera on skin.** (A) Heatmap representing the abundance variation for the most prevalent phyla on skin microbiota for the two locations (detection threshold = 0.1%, prevalence > 1%; data transformed with clr for samples, followed by scaling to zero mean, unit variance on the taxonomic features); (B) Location-wise relative abundances of the most prevalent genera (detection threshold = 0.1%, prevalence > 1%) in families. Ahmednagar (*n* = 5) and Pune (*n* = 10) The "Other" represents the sum of relative abundances for the less prevalent genera. The samples are ordered based on Staphylococcus abundance. (C) Differential abundance analysis of *Staphylococcus* (Wilcoxon test, *p* = 0.007) *Bacillus* (Wilcoxon test, *p* = 0.004) across two geographical locations with the ancombc2 method (1% prevalence).

**Table 5  Family-specific skin microbiome variation of significant genera.** The p-values calculated based on the Dunn test using the Kruskal-Wallis test and adjusted with the "Benjamini–Hochberg (BH)" method. p-values shown only for significant family comparisons.

| Family comparison | Geographical location of families | Genera showing quantitative differences | p-value |
|---|---|---|---|
| A-C | Ahmednagar-Pune | *Bacillus* | 0.02* |
| A-F | Ahmednagar-Pune | *Bacillus* | 0.01** |
| A-L | Ahmednagar-Pune | *Bacillus* | 0.04* |
| A-N | Ahmednagar-Pune | *Bacillus* | 0.01** |
| B-D | Ahmednagar-Ahmednagar | *Anaerococuss* | 0.01** |
| B-E | Ahmednagar-Ahmednagar | *Anaerococuss* | 0.01** |
| B-C | Ahmednagar-Pune | *Bacillus* | 0.05* |
| B-F | Ahmednagar-Pune | *Bacillus* | 0.02* |
| B-H | Ahmednagar-Pune | *Staphylococcus* | 0.05* |
| B-H | Ahmednagar-Pune | *Anaerococuss* | 0.02* |
| C-E | Pune-Ahmednagar | *Bacillus* | 0.01** |
| C-K | Pune-Ahmednagar | *Bacillus* | 0.01** |
| D-H | Ahmednagar-Pune | *Staphylococcus* | 0.04* |
| D-M | Ahmednagar-Pune | *Anaerococuss* | 0.04* |
| E-F | Ahmednagar-Pune | *Bacillus* | 0.01** |
| E-H | Ahmednagar-Pune | *Staphylococcus* | 0.02* |
| E-I | Ahmednagar-Pune | *Bacillus* | 0.02* |
| E-I | Ahmednagar-Pune | *Staphylococcus* | 0.05* |
| E-L | Ahmednagar-Pune | *Bacillus* | 0.01** |
| E-L | Ahmednagar-Pune | *Anaerococuss* | 0.05* |
| E-M | Ahmednagar-Pune | *Anaerococuss* | 0.03* |
| E-N | Ahmednagar-Pune | *Bacillus* | 0.00**** |
| E-O | Ahmednagar-Pune | *Bacillus* | 0.01** |
| F-K | Pune-Ahmednagar | *Bacillus* | 0.02* |
| F-M | Pune-Ahmednagar | *Anaerococuss* | 0.05* |
| K-N | Ahmednagar-Pune | *Bacillus* | 0.01** |

**Note:**
The p-value indicates level of significance <0.05*, <0.01**, <0.00****.

(Kruskal-Wallis test, FDR < 0.1, Fig. 4F). This parent-child analysis was conducted on the thirteen families that included both parents and children regardless of their sex. We excluded two families from Ahmednagar due to the death of one of the parents in G2.

## DISCUSSION

The composition of skin microbiota is diverse and is influenced by multiple factors. Several studies have indicated the predominance of the top four phyla-*Actinobacteria*, *Proteobacteria*, *Firmicutes*, and *Bacteroidetes* on healthy human skin (*Grice et al., 2009*; *Pammi et al., 2017*; *Schoch et al., 2019*). Likewise, in the present study, we report the prevalence of phyla *Firmicutes* followed by *Proteobacteria* and *Actinobacteria* on the skin of family members. Our results are consistent with earlier studies on the skin microbiota of Indian families, which also revealed an abundance of *Firmicutes, Proteobacteria*, and *Actinobacteria* (*Chaudhari et al., 2020*). Similarly, our present results align with our

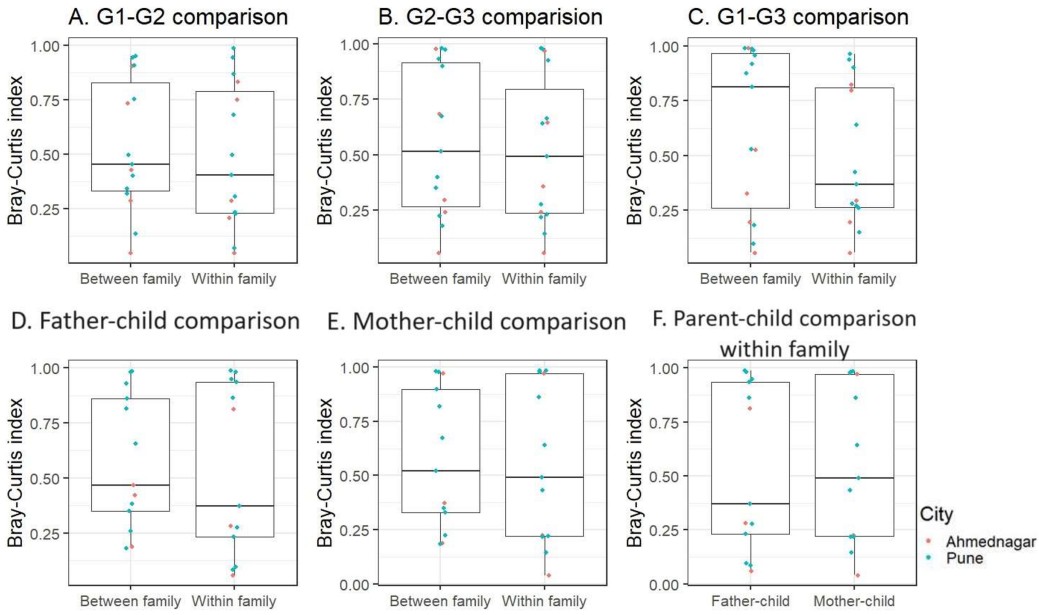

**Figure 4 The skin microbiota comparison for within and between family members across three generations.** (A) G1–G2 (grandparent-parent); (B) G2–G3 (parent-children); (C) G1–G3 (grandparent-child). Separate parent-child comparison (G2–G3) for within and between family; (D) father-child; (E) mother-child; (F) parent-child comparison within family. This is compared to the baseline dissimilarity in random adult-child pairs drawn from different families. The differences were not significant (FDR < 0.1, Kruskal-Wallis test) for all comparisons.

previous findings on the skin microbiota of genetically unrelated individuals, wherein we reported a predominance of the same phyla-*Firmicutes, Proteobacteria*, and *Actinobacteria* (*Potbhare et al., 2022*). A study from the United States on infants' skin microbiota has reported the dominance of *Firmicutes*, followed by an abundance of *Actinobacteria*, and *Proteobacteria* (*Roux, Oddos & Stamatas, 2022*).

Further, analysis of core taxonomic composition revealed the prevalence of commonly colonized skin genera like *Staphylococcus, Bacillus, Pseudomonas, Anaerococcus, Corynebacterium*, and *Dermabacter* in the study population. However, relative abundances of core genera varied from a previously reported study by *Roux, Oddos & Stamatas (2022)*, where the predominance of *Streptococcus* (52.8%), *Cutibacterium* (11.8%), and *Staphylococcus* (8.1%) in infant skin microbiota has been suggested (*Roux, Oddos & Stamatas, 2022*). Likewise, various studies demonstrated heterogeneous distribution and dominance of *Staphylococcus, Corynebacterium, Streptococcus,* and *Cutibacterium* genera on skin sites (*Saville et al., 2022*; *Zeeuwen et al., 2012*; *Ederveen et al., 2020*; *Byrd, Belkaid & Segre, 2018*). Studies based on skin type reported that *Staphylococcus* is dominant in the sebaceous and moist type, while *Propionibacterium* dominates in the sebaceous and *Corynebacterium* only in the moist (*Catinean et al., 2019*) skin type however, irrespective of skin type and site, *Cutibacterium, Staphylococcus,* and *Corynebacterium* are present and account for about 45% to 80% of the total skin microbiota (*Samaras & Hoptroff, 2020*).

## Skin microbiota association for various confounding factors

Nutrition is one of the most crucial parameters in modulating skin health and condition. Several studies have been conducted on the gut to understand the influence of diet on microbiota (*Bibbo et al., 2016*; *Schoeler & Caesar, 2019*); however, no direct link has been established between skin microbiota composition and diet. We observed a borderline association in alpha diversity of vegetarian and mixed (consuming both the plant and animal-based food items, such as eggs, meat, seafood, and poultry) diets; however, beta diversity did not show significant differences between the two diet groups.

Likewise, our present study found no significant age-related differences in microbiota diversity (either alpha or beta) among individuals of three generations across families. Our results align with the study of *Chaudhari et al. (2020)*, where three generations of Indian patrilineal families were studied to investigate the association of age on skin microbiota. Their comparative analysis revealed no significant differences in alpha diversity of the skin and gut microbiota with three age groups ranging from 3 to >50 years (*Chaudhari et al., 2020*). Because these studies utilize genetically related individuals from Indian families spanning different age ranges, the skin microbiota similarities among the three generations' age groups could be linked to co-habitation or genetic relatedness. However, neither of these studies involves the longitudinal sampling of individuals; and, microbiota shifts over a period of time have not been investigated. However, recent studies by *Seo et al. (2023)*, showed an increased skin bacterial diversity when individuals were compared after 6 years.

Additionally, gender-specific differences in skin bacteria depend on numerous factors such as skin topography and physiology, *e.g.*, skin thickness, pH, number of hairs, distribution of sweat glands, use of cosmetics, personal hygiene, hormone levels, *etc.*, (*Ruuskanen et al., 2022*; *Skowron et al., 2021*). A study by *Oh et al. (2012)* demonstrated no gender-specific significant association on the skin microbiota of the volar forearm. Likewise, we did not observe sex-specific differences in skin microbiota. This could again be due to cohabitation, minimising the effect of sex. A recent study by *Pagac, Stalder & Campiche (2024)* reported that the menstrual cycle influences skin microbiota composition when pre- and post-menopausal females were compared. It was also evident that women with irregular menstrual cycles had significantly lower skin microbiota diversity.

These confounding factors are currently overrepresented in human microbiome research of U.S and other western nations limiting the practicality of the findings to the populations with diverse genetic background, ethnicity, environment, and lifestyle (*Grice & Julia, 2011*; *Smythe & Wilkinson, 2023*). This highlights the need for further research on skin microbiome diversity and potential confounders to gain a comprehensive understanding at global level especially the continents that are often overlooked like Asia, Africa *etc*. Therefore, the present study contributes to the efforts to characterise the human skin microbiome in diverse Indian populations.

## Skin microbiota association with geographical location

The spatial abiotic factors or climatic conditions are directly related to the host as they affect skin bacterial taxonomic composition; therefore, the microbiota composition of an

individual can provide spatial information about a person (*Ruuskanen et al., 2021*). In our study, a borderline association with alpha diversity and a significant association with beta diversity has been observed in the skin microbiota of individuals living in two different geographical locations. We also reported that the families of different geographical locations significantly vary regarding the relative abundance of two core genera, *staphylococcus* and *Bacillus*. Further analysis based on twenty-five genera, using differential abundance analysis (DAA), strengthens our finding that skin microbiota varies significantly with geographical location. The selected geographical locations vary in climatic conditions and urbanization. Geographically, Pune covers an area of about 1,110 km$^2$ and has a dry climate with an average temperature ranging from 20 °C to 28 °C. Ahmednagar has a hot and semi-arid climate with an average temperature of 15 °C to 37 °C and receives lower rainfall (~46.3 mm) than Pune (~67.9 mm). A study by *McCall et al. (2020)* explained the effect of urbanization on skin microbial compositions by evaluating house structures, outdoor exposure, and the number of inhabitants of different villages in Amazone. Urbanization represents a modern lifestyle, good amenities, increased exposure to chemicals, pollution, hygiene, personal care routines, *etc*. Pune is a metropolitan city, highly urbanized by modern lifestyle, educational development, technology, industries, human resources, health facilities, and transportation system and other differences, compared to Ahmednagar, where the level of urbanization is low. In Ahmednagar, occupationally, people are mostly engaged in farming, animal husbandry practices, *etc*., hence they are more exposed to the environment, which might be the cause of differences in their skin bacterial communities. *Ying et al. (2015)* demonstrated that skin bacteria are significantly influenced by urban and rural life-style.

## Skin microbiota variations in family members

An earlier microbiota study by *Ross, Doxey & Neufeld (2017)* reported that cohabiting couples shared more similar foot skin microbiota. *Song et al. (2013)* compared 60 families with 36 dogs and found that co-habitation resulted in skin microbiota similarities for all comparisons, including human-human, dog-dog, and human-dog. They further demonstrated that members of the same family shared similar skin microbiota to members of different households, stating that a shared environment may homogenize skin microbiota through contact with common surfaces (*Song et al., 2013*). In the present study, we did not observe the significant differences in 'within' and 'between' family comparisons across three generations (G1–G2, G1–G3, G2–G3), however, it was evident that family members share on an average similar microbiome that non-family members. Overall, we observed that the Bray-Curtis difference had intermediate level (0.25–0.75) when members within same family compared across three generations. But in certain cases, even family members exhibited high differences in skin microbiota composition (BCI >= 0.75). This could be due to the outdoor activities or gender or co-habitation as the diet, age and geographical location of the family members were same. Also, these differences may attribute due to individuals' life style factors, food preferences, its cooking style, and clothing.

Consequently, it raises the question of whether skin microbiota carries a distinctive familial signature. However, drawing such a conclusion would necessitate a dedicated study with a larger sample size and a more significant number of individuals within the same generation. Similar to what has been suggested previously for gut microbiota, genetically related individuals might exhibit more similarity in their gut microbiome than genetically unrelated individuals, regardless of co-habitation patterns (*Turnbaugh et al., 2009*). *Yatsunenko et al. (2012)* observed that teenagers share more similar faecal microbiota with their parents than with unrelated adults. A study by *Si et al. (2015)* on the skin microbiota diversity of monozygotic and dizygotic twins suggested that variation is due to host genetics, age, and skin pigmentations. A recent study on facial skin microbiota investigated genetic factors that influence ageing-related pathways causing skin microbial differences in healthy Italian women in three age groups (younger, middle-aged, and older) (*Russo et al., 2023*). However, an in-depth analysis of the heritability of a skin microbiota across generations will require a more significant number of families and samples per generation.

## LIMITATIONS

The objective of the current study was to compare families to identify family-specific microbiota variation and to understand the associations of genetics with skin microbiota composition between genetically related and unrelated individuals within and between families across three generations. However, one of the major limitations of the present study is that most families included had only one grandparent or a single sibling in G2. Consequently, there was variability in the number of family members present across the three generations within the enrolled families. The outcomes of this study could have been more robust had the analysis based on more individuals per generation (such as cousins) to encompass the full spectrum of genetically related and unrelated family members. Also, relatively number of individuals within a family (in G1 and G3) limited our thorough exploration of within-family variation and the impact of shared environment. To overcome this limitation, including more families is imperative to more comprehensively explore the intricate interplay between the similarity of skin microbiota, genetic relatedness, and the diverse confounders. The statistical power of this study is limited, and although we did not identify taxonomic groups unique to a family, but a significant genus-level variation between families was observed. Furthermore, longitudinal sampling of families would be necessary to address age-related changes in the skin microbiota.

## CONCLUSION

The present study highlighted the composition of skin microbiota from Indian families, which was found to be typically characterized by the presence of three phyla such as *Firmicutes*, *Proteobacteria*, and *Actinobacteria*, and the six prevalent genera *Staphylococcus*, *Bacillus*, *Pseudomonas*, *Anaerococcus*, *Corynebacterium*, and *Dermabacter*. Further, our findings quantify the relative contribution of different sources of variation by analysing the effects of diet, age, and geographical location.

We report significant differences in the abundances of *Staphylococcus* and *Bacillus* based on geographical location. Furthermore, we found significant genus-level variation between families. Further, no significant differences were observed when within and between family comparisons were made across generations. However, it was evident that family members shared, on average, a similar microbiome to members from different families. Our study highlights geographical location and family contribute significantly in shaping skin microbiome of genetically related and other individuals, whereas, cohabitation masks the effect of other confounding factors, such as diet, age, and gender. For a comprehensive understanding of the interplay between genetics and cohabitation, it is imperative to undertake future research with a significant number of families with more siblings spanning multiple generations.

## ACKNOWLEDGEMENTS

The authors are grateful to the volunteers for their participation in this study. The authors thank Mr. Manoj Vaikar for sample collection. We also thank Pyry Kantanen for help with map visualization. We are grateful to the Department of Zoology for providing infrastructure and equipment facilities.

### Funding

This work was supported by the MHRD-SPARC (Government of India) through financial support. This work was also supported by the Rashtriya Uchchtar Shiksha Abhiyan (RUSA, Savitribai Phule Pune University) through a student fellowship. The funders had no role in study design, data collection and analysis, decision to publish, or preparation of the manuscript.

### Grant Disclosures

The following grant information was disclosed by the authors:
MHRD-SPARC (Government of India).
Rashtriya Uchchtar Shiksha Abhiyan (RUSA, Savitribai Phule Pune University).

### Competing Interests

The authors declare that they have no conflict of interests.

### Author Contributions

- Renuka Potbhare conceived and designed the experiments, performed the experiments, analyzed the data, prepared figures and/or tables, authored or reviewed drafts of the article, drafted the original manuscript, and approved the final draft.
- Ameeta RaviKumar conceived and designed the experiments, authored or reviewed drafts of the article, reviewed and edited the manuscript, and approved the final draft.
- Eveliina Munukka conceived and designed the experiments, authored or reviewed drafts of the article, reviewed and edited the manuscript, and approved the final draft.

- Leo Lahti conceived and designed the experiments, analyzed the data, authored or reviewed drafts of the article, coordinated the statistical analysis and data science, reviewed and edited the manuscript, and approved the final draft.
- Richa Ashma conceived and designed the experiments, authored or reviewed drafts of the article, reviewed and edited the manuscript, and approved the final draft.

## Human Ethics

The following information was supplied relating to ethical approvals (*i.e.*, approving body and any reference numbers):

The study was approved by an institutional human ethics committee of Savitribai Phule Pune University (Letter No. SPPU/IEC/2019/57).

## Data Availability

All raw sequences are available at European Nucleotide Archive: PRJEB44216 (ERS6234946 to ERS6234956, ERS6234959 to ERS6234976, ERS6234984 to ERS6234989) and PRJEB62887 (ERR11891529 to ERR11891565).

The source code for the analysis is available at GitHub and Zenodo:

- https://github.com/Renuka-3/Family_microbiome/tree/v1.0

- Renuka Potbhare, Ameeta Ravikumar, Eveliina Munukka, Richa Ashma, & Leo Lahti. (2023). Skin microbiota variation in Indian families (v1.0). Zenodo. https://doi.org/10.5281/zenodo.10297063.

The OTU table, Taxonomy and metadata are available in the Supplemental File.

## Supplemental Information

Supplemental information for this article can be found online at http://dx.doi.org/10.7717/peerj.18881#supplemental-information.

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
