# Peer review of "Skin microbiota variation in Indian families"

_PeerJ, doi:10.7717/peerj.18881_

## Round 0.1 · original submission · Major Revisions

Dear authors,

We kindly request that you carefully review the comments provided by the reviewers. Their valuable suggestions offer insights to enhance your manuscript. Incorporate their suggestions and carefully address all comments in your manuscript; it will significantly strengthen its content. Thanks

Reviewer 1 ·

Basic reporting

Writings need minor modifications, such as

Background in abstract (line number missing), “joint families often encompass members spanning multiple generations cohabiting in the same household, thereby sharing [the same] ethnicity, genetics, dietary habits, lifestyles, and other living conditions”.

There are some vague terms or descriptions that need to be clarified, such as

Results in abstract (line number missing), “the skin microbiota composition is typically comprised of the three most prevalent phyla, Firmicutes, Proteobacteria, and Actinobacteria”. The authors should clarify these three phyla are most prevalent in what sense (e.g., in what environmental niches).

Line 35-36, “However, depending on the skin's micro-environment, these bacteria may have a positive or opportunistic impact”. The authors should clarify what does it mean “positive or opportunistic impact”.

There are places where citations should be added, such as

Line 45-46, “babies born vaginally exhibit a microbiota composition similar to that of their mother's birth canal.”

The abstract is too long. Conventionally, the abstract should contain 1-2 sentences of background, 1-2 sentences of methods, 2-3 sentences of results, and 1 sentence of conclusion. The authors have place too many sentences in each sub-section with unnecessary method/result details. I suggest the authors shorten the text.

Experimental design

The research question is well-defined and can be summarized as: “to understand skin microbiome variation within and between families across three generations”.

The methods (sampling, bioinformatics, statistics) are quite standardized in this type of study.

However, the methodology is flawed mostly because the samples were collected only once. It is expected that the armpit/axilla microbiota may vary due to many conditions such as temperature, sweating, clothing material, etc. And that variation per se also reflects within/between family differences. Essentially, the authors simply took a snapshot of the skin microbiota and used it to draw their conclusions without taking variations or more comprehensive microbiome profile into consideration.

Validity of the findings

Although the review guideline clearly states that the impact is not part of the assessment, a scientific paper should deliver findings that help researchers further their research. In this manuscript, my major concern is that I do not see any valuable findings. The conclusion is basically that the skin microbiota vary by locations, which is the only significant association among other factors they observed. However, that is not something worth reporting, as it is a known fact that microbiota more or less vary in almost everything. Also, the variation reported was too general—there must be some phyla that list as the top n phyla, but no solid conclusion can be drawn based on a few phyla’s difference, and those few-phyla difference alone is not so meaningful.

Other minor concerns include

1. P-value = 0.05 is generally considered as the threshold of statistical significant. But the authors affirm significance at p-value=0.08 for geographical location and diet.
2. Line 403-404, “skin microbiota composition might potentially act as a forensic biomarker”. It is a very bold statement that requires supporting evidence.

Cite this review as

Reviewer 2 ·

Basic reporting

The introduction provides a solid background, but the rationale for choosing multi-generational families specifically could be further emphasized, especially regarding why this model is particularly valuable for studying skin microbiota.

Sharing of metadata related to the sequencing results could be beneficial for transparency

provide figures of better quality

Experimental design

A more explicit discussion of the limitations due to the sample size (i.e., families with only one grandparent or sibling) could be addressed earlier in the paper. This would manage expectations about the study's power and generalizability

Further detail on how potential biases during sample collection (e.g., temperature changes, collection timing) were minimized could add rigor to the methodology

Validity of the findings

How do these results contribute to global understanding, particularly given the Indian population's unique lifestyle factors?

While the authors mention the limitation of sample size, more robust discussion around the implications of not having more genetically diverse or unrelated family members within the cohort would be useful.

Additional comments

While the manuscript is well-structured and contributes to the understanding of skin microbiota in Indian families, addressing the above points regarding methodology clarity and better highlighting the findings would strengthen the paper further.

Cite this review as

·

Basic reporting

The authors have made many improvements in addressing the initial concerns regarding basic reporting. The revised manuscript features a clearer and more focused introduction, with a streamlined discussion of the study's objectives and background. The authors have also made substantial efforts to enhance the readability and organization of the manuscript, including the resolution of figures, which now meet the publication standards.

Moreover, the supplementary materials have been well integrated, with clear references in the main text. The authors have included additional explanations where necessary, making it easier for readers to navigate the supplementary content. Overall, the revisions in this section have effectively addressed the initial concerns, resulting in a well-structured and coherent manuscript.

Experimental design

The revisions to the experimental design section demonstrate that the authors have carefully considered the feedback provided by the reviewers. They have provided a justification for selecting the V3-V4 region for sequencing, highlighting its advantages in terms of resolution and accuracy for skin microbiota studies. The rationale behind the sampling strategy, particularly the focus on the axillary region, is now well-articulated, emphasizing the region's accessibility and its relevance for capturing a representative microbiota profile.

Additionally, the authors have acknowledged the limitations of their study, particularly the sample size, and have discussed how this limitation might impact the robustness of their findings. They have also enhanced the clarity of the methodology, providing more detailed descriptions of sample collection, DNA extraction, and the controls used during sequencing. These revisions have strengthened the manuscript's experimental design, making it more rigorous and transparent.

Validity of the findings

The authors have adequately addressed the concerns related to the validity of their findings. They have provided a thorough discussion of the study's results, particularly the impact of geographical location on skin microbiota composition and the genera-specific variations observed across families. The authors have effectively used differential abundance analysis to support their findings and have discussed the potential ecological and functional implications of these variations.

The revisions also include a more nuanced interpretation of the alpha and beta diversity metrics, placing their findings within the context of existing literature on skin microbiota. While acknowledging the study's limitations, the authors have provided a balanced discussion that underscores the significance of their results while also suggesting areas for future research. The manuscript's conclusions are now well-supported by the data, and the overall validity of the findings is strong.

Additional comments

The authors have addressed many of the concerns across all sections of the manuscript. The revisions have improved the clarity, rigor, and overall quality of the study, making it suitable for publication.

·

Basic reporting

Potbhare et al. investigated the impact of geographic location, diet, age, sex on skin microbiota within and between families. Location has been identified as the most important factor controlling microbial compositions and beta diversity. At this point, PCA and RDA analyses on all samples together, would mask other factors identified for impacting the skin microbiota. I suggest to do PCA and RDA analyses on Pune and Ahmednagar samples separately, and identify significant factors (p<=0.05) for each location.

Figure 4A, 4B and 4C showed large difference within the same family members, implying factors other than location have significant impacts on skin microbiota within family. Please detect what families show the largest differences (e.g. bray-curtis distance >=0.75) within the same family (G1-G2, G2-G3, and G1-G3). For example, families A and B show the largest within-family difference between G1-G2; families A, B, and D for G2-G3; and families B and D for G1-G3. Why do families A, B and D show the largest difference? Do these families all have vegetarian as diet? Or other factors make these families distinct from the other families?

Since the difference within the same family is large, I would suggest to pair-wise compare two members from each generation within the same family, while these two members are available. This is also to determine the impact of sex on skin microbiota when diet, age and location are the same.

Lines 62-72: Skin microbiota is different from gut microbiota. The study of gut microbiota is less related to that of skin microbiota. The effects on gut microbiota are not necessarily the same ones on skin microbiota unless there is research to connect these two microbiota. Please focus on skin microbiota and its controlling factors. Delete the introduction on gut microbiota.

Lines 161-162: Please provide software used to filter sequences and their parameter setting. Filtering couldn’t remove chimeras. DADA2 and USEARCH could remove chimeras. Please clarify.

Lines 187-188: One member from each generation was selected as representative for pair-wise comparison within family similarity. Did two members from each generation have similar microbial compositions or have low Bray-Curtis distance? If two members from each generation in each family don’t show high similarities, what is the point to randomly select one as representative of the generation. Please show Bray-Curtis distance between two members from each generation within each family, if there is two members from this generation.

Lines 188-190: I am confused about how to calculate between family differences. For example G1-G2 between family, does it mean one member of G1 in family A was pair-wise compared with one member of G2 from all other families? If one member from a generation was selected as representative, the problem would be the same as Lines 187-188.

Lines 233-235: Please provide p-value for the factor of family.

Line 313: Change ‘differences’ to ‘similarities’.

Lines 314-316: Change ‘hence’ to ‘and’.

Lines 361-362: In Figure 4A, 4B and 4C, the bray-curtis distance could be higher than 0.75 or even 1, meaning high differences within the same family members. Please correct this part.

Experimental design

no comments

Validity of the findings

no comments

---

## Round 0.2 · accepted · Accept

The authors highlight that the study emphasizes both the diversity and shared characteristics of skin microbiota composition within and across families. Their findings indicate that geographical location significantly influences the genus composition of skin microbiota, which is uniquely characteristic of each family and likely shaped by co-habitation. In my opinion, this article is now acceptable

Reviewer 1 ·

Basic reporting

The authors have sufficiently addressed my previous comments in the first round of review.

Experimental design

For the previous comment of mine below, I do not find the authors' response is sufficient:

"However, the methodology is flawed mostly because the samples were collected only once. It is expected that the armpit/axilla microbiota may vary due to many conditions such as temperature, sweating, clothing material, etc. And that variation per se also reflects within/between family differences. Essentially, the authors simply took a snapshot of the skin microbiota and used it to draw their conclusions without taking variations or more comprehensive microbiome profile into consideration."

If the requirements of sampling are not set more rigorously, the findings are not convincing. Specifically, the microbiome variation on subject-level is unknown, i.e., a subject's morning microbiome may be very different from evening's. Note that setting the same sampling time does not address this limitation, because the morning sampling time may not be the representative timing for the subjects. A more scientific way is to sample the same subject multiple times in a day, or even in a week, to build a "averaged" profile for the subjects. I do not find the authors' response acceptable unless they show evidence either from literature or by experiments suggesting the the within-day or within-week skin microbiome variation is negligible compared to the research objectives (location, diet, familial relationship, etc.).

Another minor concern of this particular comment is that the minor conditions for sampling are not set consistent (such as sweating, clothing, before or meal, etc.).

Validity of the findings

I do not find the findings valid because:

1. The study aims to investigate whether the microbiome variation exists on several levels including location and family, but the sampling procedures do not ensure across-subject consistency. Imagine one subject just finished 2 hours of working in the field before sample collection, while another subject just woke up in bed, is it fair to compare these two subjects? Or image one subject has been wearing a 5-day old shirt without washing, while another subject has been top-naked overnight, is it fair to compare these two subject?

We know that blood collection for patients usually should be in the morning without drinking or eating, because drinking/eating/exercise would confound the results. In this study's scenario, many factors such as sweating/clothing/etc. could confound the results too.

2. Statistical significance is generally considered p-value<0.05. The authors should not change this standard to 0.1 simply by giving it a new name.

Cite this review as

Reviewer 2 ·

Basic reporting

no comment

Experimental design

no comment

Validity of the findings

Authors should mention the limitations of sampling sizes in the conclusions

Cite this review as

·

Basic reporting

Authors have addressed all the previous concerns in basic reporting.

Experimental design

Authors have addressed all the previous concerns in experimental design sections of the manuscript.

Validity of the findings

Authors have addressed all the previous concerns in findings section.

Additional comments

The revisions have improved the clarity, rigor, and overall quality of the study, making it suitable for publication.

·

Basic reporting

The authors addressed my concerns.

Experimental design

No comments

Validity of the findings

No comments